# Robust perisomatic GABAergic self-innervation inhibits basket cells in the human and mouse supragranular neocortex

**Viktor Szegedi[1†]\*, Melinda Paizs[1†], Judith Baka[2], Pál Barzó[3], Gábor Molnár[2], Gabor Tamas[2], Karri Lamsa[1]\***

[1]MTA-NAP Research Group for Inhibitory Interneurons and Plasticity, Department of Physiology, Anatomy and Neuroscience, University of Szeged, Szeged, Hungary; [2]MTA-SZTE Research Group for Cortical Microcircuits, Department of Physiology, Anatomy and Neuroscience, University of Szeged, Szeged, Hungary; [3]Department of Neurosurgery, University of Szeged, Szeged, Hungary

**Abstract** Inhibitory autapses are self-innervating synaptic connections in GABAergic interneurons in the brain. Autapses in neocortical layers have not been systematically investigated, and their function in different mammalian species and specific interneuron types is poorly known. We investigated GABAergic parvalbumin-expressing basket cells (pvBCs) in layer 2/3 (L2/3) in human neocortical tissue resected in deep-brain surgery, and in mice as control. Most pvBCs showed robust $GABA_AR$-mediated self-innervation in both species, but autapses were rare in nonfast-spiking GABAergic interneurons. Light- and electron microscopy analyses revealed pvBC axons innervating their own soma and proximal dendrites. GABAergic self-inhibition conductance was similar in human and mouse pvBCs and comparable to that of synapses from pvBCs to other L2/3 neurons. Autaptic conductance prolonged somatic inhibition in pvBCs after a spike and inhibited repetitive firing. Perisomatic autaptic inhibition is common in both human and mouse pvBCs of supragranular neocortex, where they efficiently control discharge of the pvBCs.

**\*For correspondence:**
szegediv@bio.u-szeged.hu (VS);
klamsa@bio.u-szeged.hu (KL)

[†]These authors contributed equally to this work

**Competing interests:** The authors declare that no competing interests exist.

## Introduction

Autapses are synapses made by a neuron onto itself (*Bekkers, 2003*; *Deleuze et al., 2014*). Although studies on experimental animals have reported autaptic self-innervation in some inhibitory as well as in excitatory neurons in the brain, (*Karabelas and Purpura, 1980*; *Lübke et al., 1996*; *Thomson et al., 1996*; *Cobb et al., 1997*; *Pouzat and Marty, 1998*; *Bacci et al., 2003*; *Connelly and Lees, 2010*; *Karayannis et al., 2010*; *Jiang et al., 2015*; *Yin et al., 2018*), little is known about autapses in human nerve cells, and only a single study has investigated the operation of autaptic self-inhibition in the human neocortex (*Jiang et al., 2012*). Hence, how autaptic inhibitory circuits operate in the human brain compared to those in common experimental animals remains largely unknown.

A number of studies in rodents have demonstrated GABAergic autapses in neocortical deep layer fast-spiking parvalbumin-expressing basket cells (pvBCs) by showing physiological and pharmacological evidence for self-inhibition. In layer 5 of the infragranular neocortex, $GABA_AR$-mediated self-inhibition prolongs the pvBC spiking interval during sustained high-frequency firing (*Bacci and Huguenard, 2006*; *Manseau et al., 2010*). Evidence for autapses also exists in the human epileptic infragranular neocortex, where high-frequency spike bursts are associated with autaptic GABA

release from fast-spiking interneurons (*Jiang et al., 2012*). However, the operation of autaptic self-inhibition in superficial neocortical layers is unknown (*Tamás et al., 1997*).

GABAergic inhibition at the perisomatic region efficiently controls spike output (*Tremblay et al., 2016*; *Feldmeyer et al., 2018*). Hence, pvBCs are key players synchronizing neuronal network activity through their inhibitory synapses (*Cobb et al., 1995*; *Pouille and Scanziani, 2001*; *Wood et al., 2017*; *Cardin, 2018*), and altered pvBC firing is often linked to pathological network activities (*Jiang et al., 2016*; *Palop and Mucke, 2016*; *Dienel and Lewis, 2019*). In addition, perisomatic inhibition through autapses can efficiently regulate pvBC firing (*Bacci and Huguenard, 2006*; *Guo et al., 2016*; *Yilmaz et al., 2016*). Computational models can help us understand the role of pvBC self-inhibition in the generation and maintenance of cortical network activities (*Connelly, 2014*; *Guo et al., 2016*; *Yilmaz et al., 2016*), but very little is known about autapses in human brain, including their occurrence and inhibitory efficacy in distinct interneuron types.

We investigated GABAergic autapses in human and mouse supragranular layer 2/3 pvBCs and some nonfast-spiking interneurons. We show that GABA$_A$R-mediated inhibition is present in most pvBCs in both species and that perisomatic autaptic contacts in these interneurons suppress excitability and inhibit repetitive firing. Autapses are rare in nonfast-spiking GABAergic interneurons. We conclude that GABAergic interneuron autapses are a standard and cell type-specific microcircuit feature in the mammalian neocortex that mediates strong perisomatic self-inhibition of pvBCs.

## Results

We investigated autapses in 46 supragranular layer 2/3 (L2/3) pvBCs and 22 nonfast spiking interneurons (nonFSINs) in human neocortical tissue resected from frontal, temporal or occipital areas during deep brain surgery to obtain access to subcortical pathological targets (tumor, cyst, aneurysm or catheter implant). 39 basket cells were identified by their axon forming boutons around unlabeled L2/3 neurons (*Figure 1—figure supplement 1*) (*Szegedi et al., 2017*). Seven unsuccessfully visualized fast-spiking interneurons were included as putative pvBCs (*Figure 1—figure supplement 1*). For comparison, we studied pvBCs in the mouse somatosensory cortex. *Supplementary file 1* lists the interneurons and shows their immunohistochemical reaction for parvalbumin (pv) or vesicular GABA transporter (vGAT), and reports spike kinetics and the firing properties. The table provides details of the cortical human tissue material.

### Autaptic GABA$_A$R-mediated self-innervation in human supragranular pvBCs

First, we recorded cells in whole-cell current clamp using high intracellular chloride (130 mM) that makes a GABA$_A$R-mediated potential robustly depolarizing (*Figure 1ai*) (*Bacci et al., 2003*; *Bacci and Huguenard, 2006*). Under these conditions, unitary spikes (interval 10 s) in 11 of the 13 pvBCs studied triggered GABA$_A$R-mediated depolarizing potentials (5.44 ± 0.73 mV peak amplitude, p=0.380, Shapiro-Wilk normality test, n = 11), showing a 0.92 ± 0.04 ms onset delay to the preceding action potential peak (p=0.132, Shapiro-Wilk normality test, n = 11) (see *Figure 1aii*). In contrast, nonFSINs (action potential inward current width 0.886 ± 0.036 ms vs. 0.536 ± 0.024 ms in pvBCs, p=0.001, Student's t-test) showed gabazine (GBZ)-sensitive autaptic potentials in only 1 of the 14 cells studied.

We visualized recorded cells (filled with biocytin) with fluorophore-streptavidin (*Figure 1aiii*) and studied their parvalbumin immunoreactivity using confocal fluorescence microscopy (*Figure 1aiv*) (*Supplementary file 1*). In some cells with fully recovered intact soma, we investigated anatomical evidence for self-innervation after avidin-biotinylated horseradish peroxidase reaction. By using a light microscopic study of somatic area (in 50 μm-thick section) (n = 5 cells), we found apparent self-innervation showing close apposition boutons (range 3–8 per cell) formed by their labeled axons to the proximal dendrites (distance to soma = 30 μm, 11–46 μm) (median, quartiles, n = 17 in 5 cells) and on the boundaries of soma (range 2–5 per soma, n = 10 in 3 cells). Because dense peroxidase products mostly obscured the observation of close appositions on the soma (*Tamás et al., 1997*), we specifically looked for somatic autaptic junctions using electron microscopy in separate pvBCs. The analysis in two cells showing electrophysiological evidence for GABAergic autapses revealed pvBC biocytin-filled axon terminal boutons forming contacts to their own soma (1 and 4 contacts)

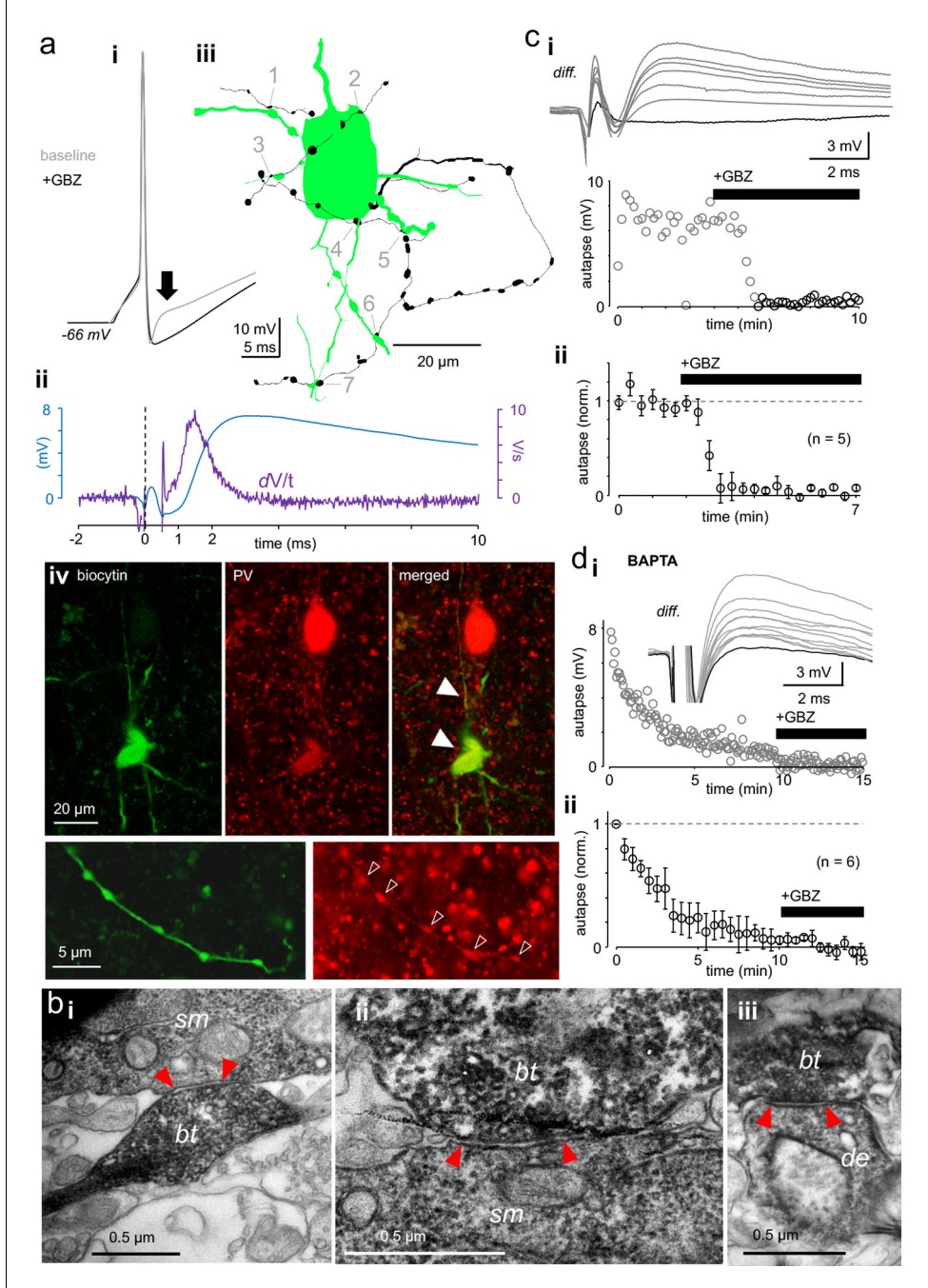

**Figure 1.** Autaptic GABA$_A$R-mediated innervation in human supragranular pvBCs. (**a**) Elevated intracellular chloride reveals GABA$_A$R-mediated self-innervation in layer 2/3 parvalbumin-immunopositive basket cells (pvBCs). (**i**) Gray trace shows unitary spike-evoked depolarizing autaptic potential (peak indicated by arrow) measured in current clamp. Black trace shows its blockage by gabazine (GBZ, 10 µM). Traces are averages of six. (**ii**) Autaptic potential (blue trace) shown as a differential of the response under control conditions and in the presence of GBZ. The first derivate (dV/t) of this potential reveals <1 ms onset delay to the action potential peak (at 0 time point). (**iii**) Light microscopic reconstruction of the same cell reveals a self-innervating axon (black), forming boutons (1-7) in close apposition to its own soma and proximal dendrites. (**iv**) Confocal fluorescence images of the same cell (biocytin-Alexa488) show pv (Cy3) immunopositive soma, dendrites (solid arrowheads) and axon (open arrowheads). (**b**) Innervation pattern of autaptic boutons. Electron microscopic images illustrate a sample biocytin-filled pvBC axon forming autaptic appositions to the soma and proximal dendrite. (**i-ii**) Sample autaptic boutons (*bt*) terminating on soma (*sm*). (**iii**) A bouton forming autaptic contact in the same cell to proximal dendrite (*de*).
*Figure 1 continued on next page*

*Figure 1 continued*

Red arrows indicate the active zone. (**c**) Autaptic response is systematically blocked by gabazine (GBZ, 10 µM). (**i**) *Top*: Unitary spike-evoked autaptic potentials shown as differential as in a_ii (*diff.*). Traces are during baseline and show the effect of GBZ wash-in (interval 10 s). Black trace shows full blockade. *Bottom*: Plot shows the GBZ effect on the autaptic potential peak amplitude in the same experiment. (**ii**) Mean ± sem of 5 similar experiments (bin 30 s, amplitude baseline-normalized). (**d**) Intracellular BAPTA (10 mM) abolishes autaptic potential gradually. (**i**) Slow inhibition of a spike-evoked autaptic potential peak amplitude in one experiment (interval 5 s). GBZ (10 µM) was applied at the end. *Inset*: Traces show vanishing autaptic potential (*diff.*) by course of experiment with BAPTA-containing filling solution. The lowest trace (full blockade) is in the presence of GBZ. (**ii**) Mean ± sem of 6 similar experiments (30 s bin, amplitude normalized by the average of the first 30 s).

The online version of this article includes the following source data and figure supplement(s) for figure 1:

**Source data 1.** Source data for *Figure 1C, D*.
**Figure supplement 1.** Identification of human L2/3 fast-spiking basket cells.

and proximal dendrites (1 and 2 contacts), revealing postautaptic density in the neuron (*Figure 1bi–iii*).

The autaptic response was readily blocked by GBZ (10 µM) (p<0.001, n = 5) (*Figure 1ci, ii*) or gradually abolished by intracellular calcium-chelator BAPTA (20 mM), which suppresses action potential-dependent vesicle release (p<0.001, n = 6) (Wilcoxon signed rank test) (*Figure 1di, ii*) (*Bacci et al., 2003*). The autaptic response amplitude and onset delay values were measured by subtracting spike-elicited responses under control conditions and in the presence of GABA_AR blocker GBZ (see *Figure 1ai–ii*).

## GABAergic self-inhibition strength in human and mouse pvBCs evoked by unitary spikes

Next, we studied autaptic self-innervation conductance (G_aut) and its inhibitory strength in pvBCs. We used voltage clamp with close-to-physiological intracellular (8 mM) chloride (*Verheugen et al., 1999*; *Connelly and Lees, 2010*) to minimize error in GABA_AR-mediated conductance emerging from artificial transmembrane chloride gradient (*Hille, 2001*). Spike-evoked autaptic outward current (interval 10 s, recorded with a holding potential between −43 and −55 mV) was uncovered by subtracting response in gabazine (average of 6 after wash-in of GBZ, 10 µM) from baseline responses (*Figure 2ai–iii*). In human pvBCs, a spike-evoked autaptic response showed peak G_aut of 3.79 nS, 2.39–6.45 nS (median, quartiles, n = 14 cells) and 4.18 ± 0.31 ms decay time constant (p=0.829, Shapiro-Wilk normality test, n = 13 cells, monoexponential curve fitting $r^2$ range from 0.86 to 0.97. A pvBC with smallest G_aut amplitude was excluded from the decay tau analysis). The autaptic current peak amplitude had a 2.31 ± 0.16 ms (p=0.066, Shapiro-Wilk normality test, n = 13) delay to the action potential inward current peak.

Altogether, experiments including the current- and voltage clamp measurements above confirmed GBZ-sensitive autapses in 25 of 36 (69.4 %) human pvBCs studied (see *Figure 2aii*), showing no difference in patient age between pvBCs with or without autapses (two-sample Kolmogorov-Smirnov test, D = 0.200 with p=0.884). There was no correlation between autapse peak conductance and patient age (Spearman's r = −0.142, p=0.615, n = 14).

On average, the human pvBC autapse peak conductance was not different from synaptic conductance (G_syn) tested in 20 monosynaptically connected pvBC-PC pairs (*Figure 2—figure supplement 1*), and it was comparable to monosynaptic conductance between pvBCs (average synaptic conductance in BC-BC three pairs; 1.04 nS, 1.68 nS and 1.84 nS). Synaptic GABA_AR-mediated peak conductance was 3.10 nS, 1.42–3.94 nS (median, quartiles) (p=0.128, Mann–Whitney U -test compared to G_aut in humans). The monosynaptic cell pairs are listed in *Supplementary file 1* with details on their spike kinetics and immunoreactivity.

Because autaptic function has not been studied in nonhuman supragranular layer interneurons, we investigated L2/3 pvBCs in the mouse somatosensory cortex. We utilized mice expressing td-Tomato fluorophore preferably in parvalbumin GABAergic neurons (*Chattopadhyaya et al., 2004*), confirmed each studied cell as fast-spiking (action inward current width 0.53 ± 0.03 ms, p=0.081, Shapiro-Wilk normality test, n = 19) (*Supplementary file 1*), and visualized cells with fluorophore-streptavidin after filling with biocytin. Three cells were further selected for immunohistochemical

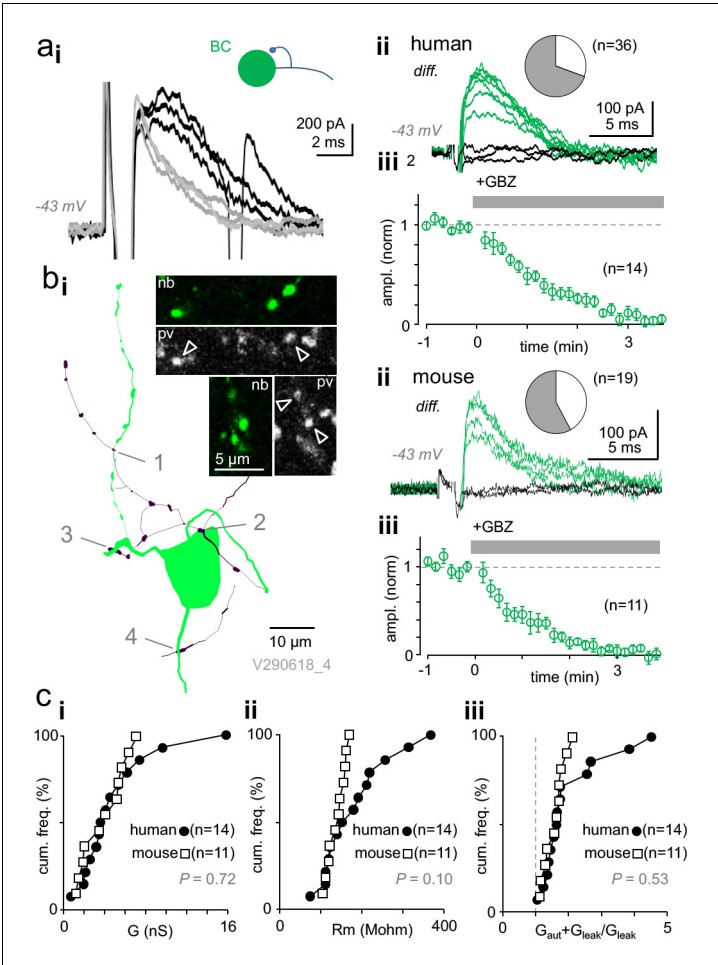

**Figure 2.** Autaptic self-inhibition strength in human and mouse pvBCs evoked by unitary spikes. (a) A voltage clamp recording in human pvBCs shows autaptic GABA$_A$R-mediated inhibitory current (intracellular chloride 8 mM). (i) Unitary spike-evoked responses in a human pvBC during baseline (black) and after wash-in of gabazine (GBZ, 5 min, gray) (interval 10 s, at −43 mV). Note the second action potential escape current generated occasionally. (ii) Subtraction of the traces reveals GABAergic current. Traces show the autaptic GBZ-sensitive current (*diff.*) in baseline (green) and when fully blocked (black, subtraction of traces in GBZ). Inset pie chart shows the proportion of cells with GBZ-sensitive autapse (green) in all pvBCs studied in voltage- or current clamp (in 25 of 36). (iii) Mean ± sem of GBZ wash-in in all pvBCs (n = 14, amplitude baseline-normalized, 30 s bin). (b) GABA$_A$R-mediated self-innervation and autaptic current in mouse pvBCs. (i) A visualized mouse pvBC perisomatic area showing axon (black) boutons forming close appositions (1-4) to its own soma and proximal dendrites. Confocal microscopic micrographs illustrate the pv immunopositive (pv, Cy5) axon of the cell (nb, Alexa488). (ii) GBZ-sensitive autaptic current (*diff.*) under control conditions (green). Black traces are the subtraction of traces in GBZ. Inset pie chart shows the proportion of mouse pvBCs with autapses (in 11 of 19 cells). (iii) Mean ± sem of 11 experiments showing GABAergic current blockade by wash-in of GBZ (amplitude baseline-normalized). (c) Autaptic self-inhibition efficacy in human and mouse pvBCs. Cumulative presentation of solid and open symbols indicates the average in cells. (i) Self-inhibition peak conductance (G) in human and mouse pvBCs. (ii) Input resistance (R$_m$) in humans and mice. (iii) Calculated total input conductance of a cell during the autaptic current peak (G$_{aut}$ + G$_{leak}$), divided by its resting conductance (G$_{leak}$). The ratio shows how much self-inhibition increases membrane leakage.

The online version of this article includes the following source data and figure supplement(s) for figure 2:

**Source data 1.** Source data for *Figure 2A, B, C*.
**Figure supplement 1.** GABAergic self-innervation is comparable to synaptic connection from pvBC to L2/3 neurons.
**Figure supplement 1—source data 1.** Source data for *Figure 2—figure supplement 1B*.
*Figure 2 continued on next page*

*Figure 2 continued*

**Figure supplement 2.** Action potential afterhyperpolarization peak conductance ($G_{ahp}$) in human pvBCs shows large variation.

**Figure supplement 2—source data 1.** Source data for *Figure 2—figure supplement 2*.

investigation and confirmed to be immunopositive for pv, as illustrated in *Figure 2bi*. We found GBZ-sensitive autapses in 11 of 19 mouse pvBCs, representing 57.9% of the cells studied (*Figure 2bii, iii*). *Figure 2bi* illustrates a visualized sample mouse pvBC with its axon forming close appositions close to its own perisomatic area. Mouse pvBCs (n = 11) had $G_{aut}$ peaks of 4.09 nS, 1.93–5.67 nS, akin to what we discovered in human pvBCs (p=0.72, Mann–Whitney U -test) (*Figure 2ci*). Autaptic current decay time constant defined by monoexponential curve fitting was 3.91 ms, 3.55–6.47 ms (n = 11).

Mouse pvBC input resistance under resting conditions (139.60 ± 6.69 MΩ, p=0.566, Shapiro-Wilk normality test) was not significantly different from $R_m$ in human pvBCs (183.83 ± 22.34 MΩ, p=0.247, Shapiro-Wilk normality test, n = 14) (p=0.077, Welch's t-test), but compared to mouse pvBCs, human pvBCs showed a wide $R_m$ value range (D = 0.500, p=0.060, two-sample Kolmogorov–Smirnov test) (see *Figure 2cii*). On average, human and mouse pvBCs exhibited comparable $G_{aut+leak}/G_{leak}$ values (p=0.985 Shapiro-Wilk normality test; p=0.358 vs. human, Student's t-test). This value shows the peak $G_{aut}$ effect on total cell conductance and reflects its capacity to reduce cell excitability. ($G_{leak}$ = membrane leak conductance in resting conditions; $G_{aut+leak}$ = total cell input conductance during peak $G_{aut}$). The $G_{aut}$ relation to $G_{leak}$ is illustrated in *Figure 2ciii*.

## Temporal window for autaptic shunting inhibition in pvBCs following a spike

The inhibitory efficacy of GABA$_A$R-mediated self-inhibition depends on $G_{aut}$ and is related to the cell $R_m$ and the relative proportion of $G_{aut}$ and intrinsic voltage-activated spike afterhyperpolarization (AHP) potassium conductance ($G_{ahp}$). We next investigated the relative strength of $G_{ahp}$ and $G_{aut}$ in pvBCs. The average $G_{ahp}$ and $G_{aut}$ traces fitted to monoexponential decay curves from their peak in one cell are illustrated in *Figure 3ai*.

Human pvBCs showed a peak $G_{ahp}$ of 17.48 ± 1.81 nS (p=0.445, Shapiro-Wilk normality test) measured from the AHP current peak in the presence of GBZ (peak at 0.78 ± 0.06 ms delay to the action inward current peak, p=0.066, Shapiro-Wilk normality test) (n = 14). The AHP outward current decay time constant from the peak was 5.43 ± 0.54 ms (p=0.190, Shapiro-Wilk normality test, n = 14). $G_{ahp}$ and $G_{aut}$ peak value (mean or median), average peak delay to action potential inward current, and monoexponential decay of $G_{ahp}$ and $G_{aut}$ is illustrated in *Figure 3aii* (n = 13, one pvBCs was excluded because monoexponential decay could not be fitted with high confidence to smallest $G_{aut}$). The *Figure 3aii* demonstrates that although $G_{ahp}$ peak amplitude was larger than $G_{aut}$ peak, the $G_{ahp}$ vs. $G_{aut}$ amplitude -ratio shows large variation between individual pvBCs.

The variability in the $G_{ahp}$ peak amplitude between individual pvBCs probably reflects the cell soma size since the peak $G_{ahp}$ value correlated with the cell capacitance ($C_m$ = 43.81, 27.65–58.60 pF, n = 14) (Spearman's r = 0.644, p=0.012, n = 14) and $G_{leak}$ (Pearson's r = 0.671, p=0.009, n = 14). *Figure 2—figure supplement 2* illustrates the correlation between $G_{ahp}$ peak amplitude and $G_{leak}$ in pvBCs. However, $C_m$ or $G_{leak}$ failed to show correlation with peak $G_{aut}$ (Spearman's test) (see *Supplementary file 2*).

The mouse $G_{ahp}$ peak amplitude was 25.21 ± 2.60 nS (p=0.096, n = 11) and the peak showed 0.59 ± 0.068 ms (p=0.312, n = 11) delay to the action inward current peak (Shapiro-Wilk normality test). The AHP current had decay time constant of 2.91 ± 0.26 ms (p=0.109, Shapiro-Wilk normality test, n = 11). Thus, the $G_{ahp}$ peak amplitude in mouse cells was higher than that in human pvBCs (p=0.007, Student's t-test), and the AHP current in mice was shorter than that in humans (decay tau mean 2.91 ms vs. 5.43 ms, p=0.002, Student's t-test). *Figure 3aiii* illustrates the peak amplitude for $G_{ahp}$ and $G_{aut}$ (mean or median), its average delay to the action inward current, and the $G_{ahp}$ and $G_{aut}$ monoexponential decays (n = 11). In addition, *Figure 3aiii* demonstrates large variation in the $G_{ahp}$ vs. $G_{aut}$ amplitude between individual pvBCs in mouse.

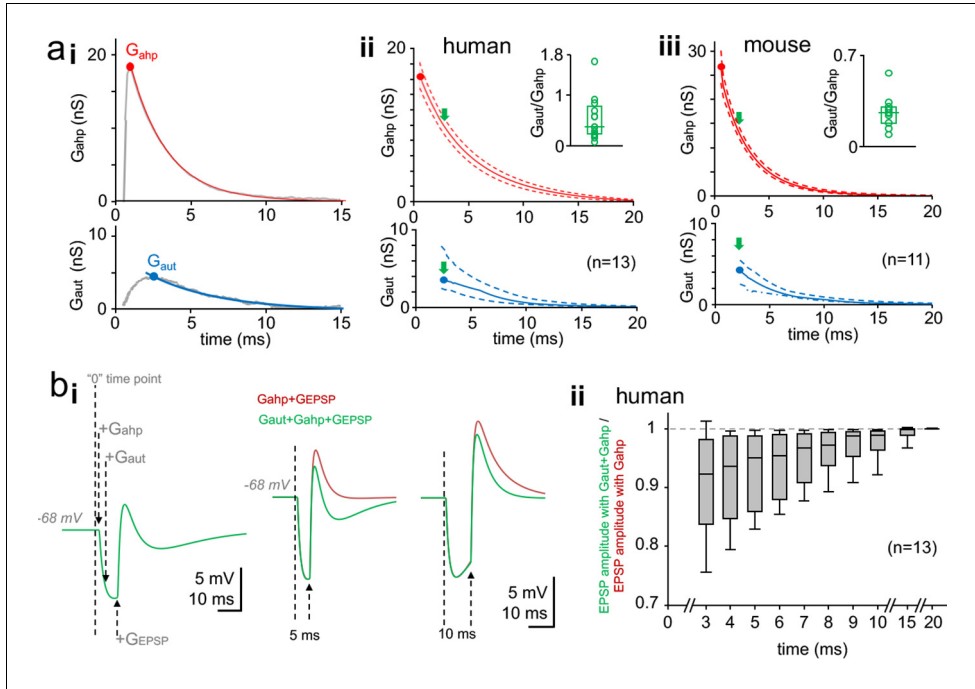

**Figure 3.** Time window for autaptic self-inhibition in pvBCs following a spike. (**a**) Autaptic GABA$_A$R-mediated conductance (G$_{aut}$) overlaps with AHP conductance (G$_{ahp}$). (**i**) The two conductances G$_{ahp}$ (top) and G$_{aut}$ (bottom) in a human pvBC (gray lines, both averages of six). Colored (top, red; bottom, blue) curves demonstrate monoexponential decay from the peak (solid circle). Abscissa '0' shows the timing of spike inward current peak. (**ii**) Monoexponential decay curves for G$_{ahp}$ and G$_{aut}$ from peak amplitude (solid circle) in human pvBCs. Red solid line shows mean and dotted lines ± sem (parametric amplitude and decay tau distribution, Shapiro-Wilk normality test). Blue solid line shows median and the dotted lines indicate upper and lower quartiles (non-parametric data distribution) (n = 13, a pvBC with smallest G$_{aut}$ excluded). Abscissa indicates delay to a spike inward current peak. Inset: high variance in G$_{aut}$/G$_{ahp}$ between cells (circles, average in each cell; box plot, median and quartiles of the averages). The G$_{aut}$/G$_{ahp}$ plotted at G$_{aut}$ peak time point (green arrows in the main plot). (**iii**) Monoexponential decay curves for G$_{ahp}$ and G$_{aut}$ in mouse pvBCs (n = 11). Inset shows G$_{aut}$/G$_{ahp}$ variation between individual cells. (**b**) Basket cell inhibition by G$_{aut}$ following a spike. Simulation study with a single-cell perisomatic model uses parameters recorded in the human pvBCs (see *Supplementary file 2*). (**i**) Membrane potential traces from a pvBC simulation demonstrate experimental design. *Left*: Vertical dotted lines show the schematic timing of conductances applied. '0' means the time point where action potential inward current peak would occur. +G$_{ahp}$ and +G$_{aut}$ show application of these conductances with cell-specific amplitude, delay and decay time. +G$_{EPSP}$ shows initiation of EPSP (using kinetics and amplitude common in human pvBCs). *Middle*: Superimposed traces show EPSP (generated with 5 ms delay to spike) in two conditions. Green trace shows the EPSP in the presence of G$_{aut}$ and G$_{ahp}$. Brown trace shows increased EPSP amplitude without G$_{aut}$. Right: G$_{EPSP}$ applied 10 ms after spike ('0' time point) in the two conditions. Turning off G$_{aut}$ still increases EPSP amplitude. Resting membrane potential (E$_m$) = −68 mV. AHP reversal potential = −90 mV. E$_{GABA-A}$ = 10.84 ± 0.53 mV negative to E$_m$. (**ii**) Summary of the simulation in 13 human pvBCs. EPSP was evoked in the two conditions with increasing delay to spike. Box plot shows 'the green trace EPSP' amplitude divided by 'the brown trace EPSP' amplitude at time points with increasing delay to spike (median, upper and lower quartiles with 5 and 95 percentiles).

The online version of this article includes the following source data for figure 3:

**Source data 1.** Source data for *Figure 3A, B*.

To study the time window for efficient autaptic inhibition in human pvBCs, we utilized a computational single-cell perisomatic model using parameters (cell-specific G$_{leak}$ and C$_m$) measured in the cells above (see *Supplementary file 2*) (*Connelly and Lees, 2010*). We defined a current input model to simulate AHP, autapse and an incoming EPSP (*Schmidt-Hieber et al., 2007*). For G$_{ahp}$ and G$_{aut}$ the time-to-peak, decay tau and the peak amplitude values were derived from the experimental data for each cell (*Supplementary file 2*). G$_{ahp}$ reversal potential was −90 mV and E$_{GABA-A}$ = −78.89 ± 0.53 mV (p=0.370, Shapiro-Wilk normality test). Resting membrane potential (Em) was

set at −68 mV by defining it as the reversal potential of $G_{leak}$. We simulated excitatory postsynaptic potential (EPSP) using glutamatergic EPSP conductance ($G_{EPSP}$) parameters of human pvBCs (rise tau 0.2 ms, decay tau 1.2 ms, conductance 10 nS, reversal potential at 0 mV) (see *Szegedi et al., 2016*). Action potential inward current component, which served as '0' time point to $G_{ahp}$, $G_{aut}$ and $G_{EPSP}$, was not included in the model since simulations focused on inhibition a few milliseconds after the spike. We applied $G_{EPSP}$ with incremental delay (from 3 to 20 ms) to $G_{ahp}$ onset (the onset in original recordings starts when action inward current ends) (see *Supplementary file 2*). First, we ran simulations in each cell with $G_{EPSP}$, $G_{ahp}$ and $G_{aut}$ (*Figure 3bi*). Next, we reproduced the simulation without $G_{aut}$ (*Figure 3bi*). *Figure 3bii* summarizes autaptic inhibition of EPSP in the 13 human pvBCs. The results demonstrate effective self-inhibition of the EPSP amplitude by GABA$_A$ receptor-mediated autaptic conductance for ten milliseconds following a spike.

## Dynamic clamp reveals efficient GABA$_A$R-mediated self-inhibition in human pvBCs

Next, we used whole cell dynamic clamp to generate somatic EPSPs in human L2/3 interneurons to study GABA$_A$R-mediated self-inhibition in pvBCs. Dynamic clamp setting is well suited to investigate autaptic inhibition because it allows generation of EPSPs in pvBCs without possible disynaptic inhibition from neighboring interneurons (*Molnár et al., 2008*; *Komlósi et al., 2012*; *Szegedi et al., 2016*).

We applied EPSCs with conductance and kinetic features (decay time constant 1.25 ms) measured in pvBCs (n = 7 cells, resting membrane potential from −63 to −78 mV). In addition to a brief (0.5 ms, up to 20 nS) suprathreshold depolarizing step, we evoked two subthreshold EPSPs (with 1.5–8 nS, amplitude 2–9 mV), one preceding 40 ms the spike (prespike EPSP) and another time-locked to elicited action potential with a 5 ms delay (postspike EPSP). Cycle interval was 15 s. Following baseline, wash-in of GBZ (10 μM) selectively increased the postspike EPSP amplitude in pvBCs to 119.6 ± 3.3% from baseline (mean of means ± sem, p=0.0001, Student's *t*-test, n = 7 cells) (P of data points in individual experiments from 0.127 to 0.995, Shapiro-Wilk normality test) that had no effect on pre-spike EPSP amplitude (102.0 ± 1.6% of baseline, n = 7). The GBZ effect in pvBCs is illustrated in *Figure 4ai–iii*. Similar experiments in nonFSINs (action inward current width 0.850 ± 0.041 ms in non-FSINs vs. 0.600 ± 0.031 in pvBCs, p=000193, *t*-test) failed to increase the prespike EPSP (amplitude 104.0 ± 1.5% of baseline) or postspike EPSP (amplitude 97.1 ± 4.0% of baseline) (n = 8). GBZ wash-in (5 min) in nonFSINs is shown in *Figure 4bi–iii*. The results confirm effective GABA$_A$R-mediated autaptic inhibition in human pvBCs.

## Autapses inhibit repetitive spike firing in human pvBCs

Finally, we used dynamic clamp to test whether somatic self-inhibition is sufficient to control pvBC action potential firing. By applying suprathreshold excitatory inward current conductance (8–21 nS, decay time constant 4–5 ms), we elicited firing of action potential doublets in three fast-spiking pvBCs studied separately and in the presence of GBZ (spike interval 6.06 ms, 5.62–6.58 ms, n = 198 doublets, 3 cells). First, we found that wash-in of GBZ shortened the spike doublet interval (*Figure 5ai–ii*), indicating GABA$_A$R-mediated inhibition of the 2$^{nd}$ spike initiation in baseline conditions. In the continuous presence of applied GBZ, in addition to the spike doublet-triggering EPSC conductance, inhibitory conductance with onset delay (1 ms to 1$^{st}$ spike), amplitude (1–10 nS) and decay time (5 ms) akin to autaptic GABA$_A$R-mediated responses were generated in dynamic clamp in pvBCs (*Figure 5bi*). Akin to simulation experiments above, resting membrane potential (range from −72 mV to −77 mV, see *Figure 5b–d*) of the cells was close to E$_{GABA-A}$ (−78 mV).

We found that by the autaptic inhibitory conductance, the doublet spike interval was elongated (*Figure 5bii*,d) (ANOVA on ranks with Dunn's post hoc test) or the 2$^{nd}$ spike probability was reduced (*Figure 5bii*, c-d). Applying 2.5 nS-10 nS autaptic inhibition by dynamic clamp reduced the 2$^{nd}$ spike probability in the three experiments to 0.34, 0.02–0.67 from 0.93, 0.69–1.00 in 0 nS autapse conditions (p=0.005, spike or no spike vs. 0 nS or 5–10 nS autapse, Chi-square test). Furthermore, as low as 1 nS autapse (tested in two of the cells) was capable of reducing 2$^{nd}$ spike initiation (*Figure 5c*). Thus, the results demonstrate that autaptic conductance measured in human pvBCs is sufficient to control spiking of the cells.

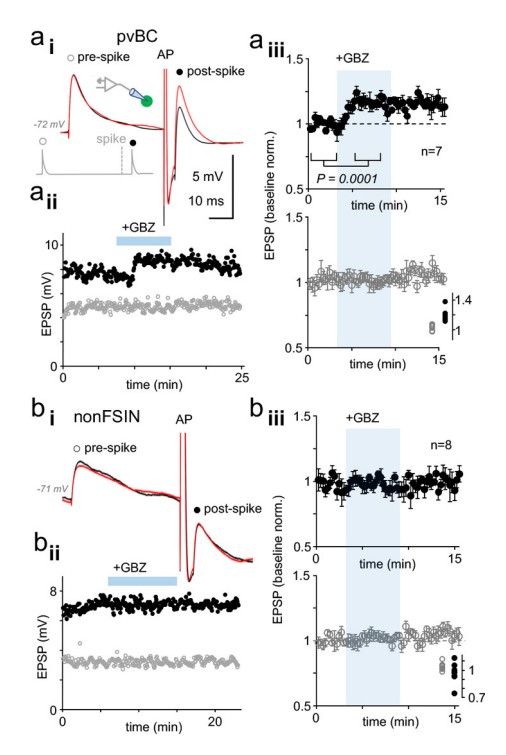

**Figure 4.** Dynamic clamp reveals autaptic inhibition of EPSPs in human pvBCs but not in nonFSINs. (a) GABA$_A$R-mediated self-inhibition of EPSP in pvBCs. A dynamic clamp experiment with two subthreshold EPSPs generated 40 ms before (prespike) and 5 ms after (postspike) a spike in pvBCs. Spike was evoked by a brief (0.5 ms) suprathreshold step. (i) Sample EPSPs (average of 6) in baseline conditions (black) and after wash-in of GBZ (10 μM) (red). Note the selective increase of the post-EPSP amplitude by GBZ. Inset schematic shows experimental design with prespike (open symbol), postspike (solid symbol) and suprathreshold (middle) conductance commands. The initiation of postspike EPSC conductance was time-locked to the peak of action potential evoked. (ii) Amplitude of the pre- (open symbols) and postspike EPSPs (solid symbols) in the same experiment (amplitude from onset to peak). GBZ wash-in is indicated by a horizontal bar. (iii) Mean ± sem of 7 pvBCs (baseline-normalized). The shaded area indicates the wash-in of GBZ. Inset summarizes the baseline-normalized pre- and postspike amplitudes in the presence of GBZ in individual experiments. (b) NonFSINs fail to show GABA$_A$R-mediated self-inhibition of somatic EPSP. (i) Traces illustrate pre- and postspike EPSP (average of 6) in a nonFSIN during baseline (black) and in GBZ (10 μM) (red). (ii) Amplitude of the pre- and postspike EPSPs plotted in the same experiment. (iii) Mean ± sem of 8 nonFSINs. Inset summarizes postspike EPSP amplitudes in GBZ (baseline-normalized).

*Figure 4 continued on next page*

## Discussion

Although autaptic self-inhibitory connections have been reported in GABAergic interneurons in rodents and some other experimental animals, their existence and function in identified interneurons in the human brain have remained virtually unknown. Furthermore, studies on autapses in animal experiments have heavily focused on neurons in infragranular layers of the neocortex, while the function of autapses in the supragranular layers has remained unexplored (*Tamás et al., 1997*; *Bacci et al., 2003*; *Bacci and Huguenard, 2006*; *Connelly and Lees, 2010*; *Jiang et al., 2012*; *Jiang et al., 2015*). Therefore, it is unclear whether robust autaptic self-inhibition is a general feature in the neocortex operating in various neocortical layers and different species (including human) or whether GABAergic self-inhibition is a specialization in deep neocortical interneurons. Our study shows that GABA$_A$R-mediated self-inhibition is a regular feature of supragranular layer pvBCs in humans and mice. pvBC axon terminals self-innervate their soma and proximal dendrites, and electron microscopic investigation shows that these contacts form 'autaptic densities' in the interneurons. The regular occurrence of autapses in pvBCs in different species and different layers and areas of the neocortex, their robust inhibitory efficacy in pvBCs and their rare occurrence in nonFSINs show that self-inhibition is a common but cell type-specific microcircuit feature in the mammalian neocortex.

### Robust perisomatic self-inhibition regulates pvBC excitability and firing in distinct cortical layers and in different mammalian species

Here, our results in the supragranular layer together with other studies in the infragranular layers show that autapses efficiently control excitability in pvBCs after spikes (*Bacci and Huguenard, 2006*; *Connelly and Lees, 2010*; *Jiang et al., 2012*; *Deleuze et al., 2019*). Autapses are present in similar proportions and show comparable inhibitory efficacy in pvBCs of superficial and deep neocortical layers (see *Bacci and Huguenard, 2006*). In addition, our results here with human and mouse cells (together with earlier studies on rat neurons) confirm that autapses have similar occurrence and strength in pvBCs in rodent and human brain slices. Correspondingly, self-innervation is rare in human nonFSINs, similar to rodent nonFSINs (*Bacci et al., 2003*; *Jiang et al., 2012*). GABAergic self-inhibition conductance in human pvBCs is comparable to

*Figure 4 continued*

The online version of this article includes the following source data for figure 4:

**Source data 1.** Source data for *Figure 4A, B*.

the synaptic inhibition these interneurons exert on neighboring layer 2/3 neurons.

Autaptic terminals on pvBC are heavily perisomatic, whereas nonFSINs, when autapses are found in them, self-innervate their own dendrites (*Tamás et al., 1997*) akin to synaptic contacts made by these interneurons to other neurons (*Favuzzi et al., 2019*). Information on the self-innervation subcellular target motifs as well as knowledge on autaptic inhibition occurrence and strength in identified neurons is essential for understanding the operation of autaptic microcircuits in individual neurons and in a neuronal network. Our results in human neocortex shed more light on this, but it is worth keeping in mind that

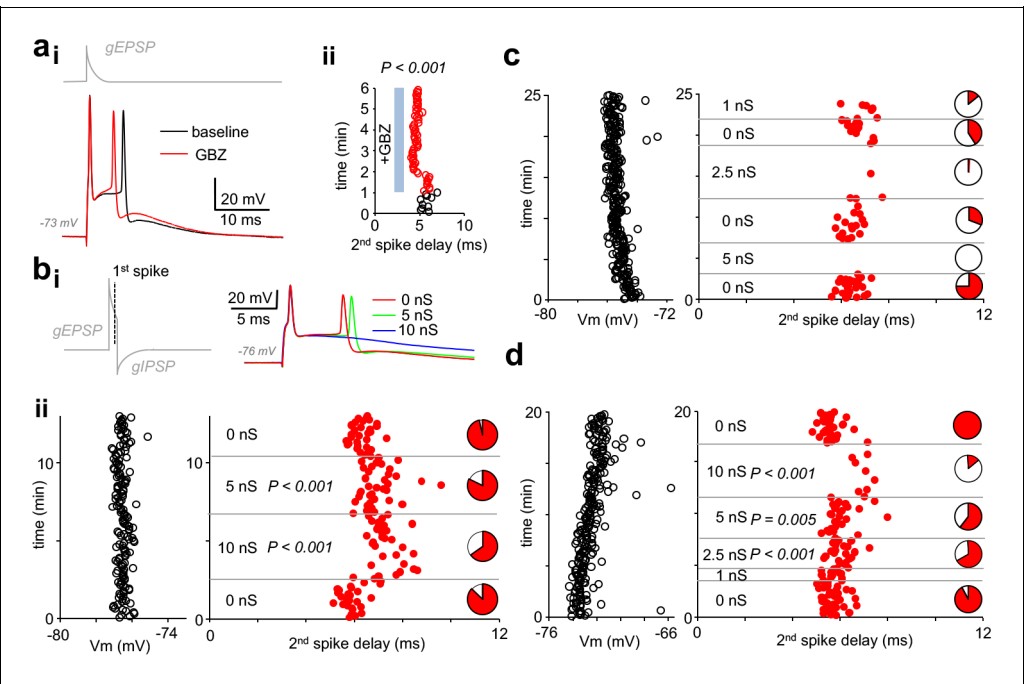

**Figure 5.** Autaptic inhibitory conductance regulates firing in human pvBCs. (a) GBZ shortens a time delay between two spikes evoked by EPSP in dynamic clamp. (i) Sample traces in a pvBC evoked by EPSP in dynamic clamp under control conditions during baseline and after wash-in of GBZ (10 μM). Trace on top (gray) shows EPSP conductance command (gEPSP). (ii) Delay from the 1st to the 2nd spike plotted in the same experiment under control conditions (black line circles) and during wash-in of GBZ (red line circles). Cycle interval 5 s. (b) Inhibition of doublet spiking by natural autaptic conductance demonstrated with dynamic clamp in human pvBCs. Experiments in the presence of GBZ show decreased probability and increased delay of the second spike when autaptic conductance is introduced with dynamic clamp. Spikes are evoked by large single EPSP as in (a). (i) *Left:* dynamic clamp command schematic with EPSP conductance followed by $GABA_A$R autaptic IPSC conductance (with 1 ms onset delay to 1st spike generated by EPSP). *Right:* traces showing the evoked spikes. (ii) *Left:* pvBC membrane potential during an experiment. *Middle:* red dots indicate the second spike onset delay to the first spike elicited by dynamic clamp EPSP (interval 5 s) in the absence of autapse, and when it was followed by autaptic IPSC conductance (5 nS or 10 nS). P-values show spike delay increased by autaptic conductance compared to baseline (ANOVA on ranks, post hoc Dunn's pairwise test against all 0 nS data as control). *Right:* pie charts show the 2nd spike probability (red area) in all cycles with the specific IPSC conductance. Note the increased failure rate (white pie chart area) with 5 nS or 10 nS inhibitory autapse. (c-d) Similar experiments as in (b) in two additional pvBCs in which autaptic inhibition (1 nS −10 nS) was more clearly seen in the probability of the 2nd spike (white areas in pie charts indicate spike failures). P-values show increased spike delay compared to baseline (ANOVA on ranks with post hoc Dunn's pairwise test against all 0 nS data as control).

The online version of this article includes the following source data for figure 5:

**Source data 1.** Source data for *Figure 5B, C, D*.

the incidence of autaptic connectivity, and hence the strength of autaptic inhibition particularly in dendrites can be influenced by resected axons of brain slices.

In pvBCs, perisomatic GABAergic self-inhibition overlaps with much stronger AHP potassium conductance. However, the autaptic conductance onset delay and late peak time make $GABA_A$R-mediated self-inhibition highly influential during AHP conductance decay. In this way, the autaptic activity strengthens somatic inhibitory conductance up to ~10 ms following a spike. We demonstrated this using a single-cell model as well as dynamic clamp experiments and showed that autaptic inhibition elongates the action potential interval (*Bacci and Huguenard, 2006*; *Connelly, 2014*; *Guo et al., 2016*; *Yilmaz et al., 2016*; *Deleuze et al., 2019*) through shunting inhibition in human pvBCs. GABAergic inhibition through shunting is particularly relevant in pvBCs since the $GABA_A$ reversal potential in these cells is often close to the resting membrane potential (*Verheugen et al., 1999*; *Connelly and Lees, 2010*). Overall, perisomatic self-innervation in human and mouse pvBCs reinforces self-inhibition after a spike. This process adjusts the pvBC firing interval and rhythmic inhibition from pvBCs to other neurons (*Guo et al., 2016*; *Yilmaz et al., 2016*; *Deleuze et al., 2019*). However, autaptic self-innervation strength shows substantial variability between individual pvBCs, and therefore it is likely that autaptic GABAergic contacts undergo activity-dependent plasticity which regulates their strength (*Castillo et al., 2011*; *Donato et al., 2013*; *Griffen and Maffei, 2014*; *Lourenço et al., 2014*; *Lourenço et al., 2019*).

## pvBC features in humans and mice

Comparison of autapses or cell input resistance showed no difference between human and mouse pvBCs. However, close investigation of the data reveals that compared to mouse pvBCs, human pvBCs exhibit a wide range of autapse conductance as well as cell input resistance. One potential explanation for this finding is human tissue material diversity. Thus, cortical region specificity and patient gender and age may partly be behind the parameter variability in human pvBCs.

However, among various cellular features characterized in human and in rodent neocortex (*Blazquez-Llorca et al., 2010*; *Testa-Silva et al., 2010*; *Molnár et al., 2016*; *Szegedi et al., 2016*; *Wang et al., 2016*; *Beaulieu-Laroche et al., 2018*; *Boldog et al., 2018*; *Goriounova et al., 2018*; *Kalmbach et al., 2018*; *Pruunsild and Bading, 2019*), human neocortical neurons often exhibit higher $R_m$ than their rodent counterparts do (*Eyal et al., 2016*; *Poorthuis et al., 2018*). Although our data here showed that human and mouse pvBC input resistance values were not different on average, human neurons had individual cells showing clearly higher $R_m$ than did those found in any of the mouse pvBCs (*Poorthuis et al., 2018*).

In line with this result, we found that AHP conductance was generally higher in mouse cells than in human cells. High $R_m$ needs less current to generate an equal amplitude potential. Indeed, human pvBC showed a correlation between AHP peak conductance and membrane leak conductance. Thus, high $R_m$ pvBCs in the human neocortex need smaller conductance for efficient inhibition than do pvBCs in mice.

## Conclusions

Firing of pvBCs coordinates synchrony of neuronal networks in the cortex (*Hu and Jonas, 2014*; *Lu et al., 2017*; *Cardin, 2018*). Through robust autapses, pvBCs adjust their temporal firing interval and correspondingly set inhibition in their target neurons. This mechanism may be essential in setting cell assembly discharges in the neocortex (*Molnár et al., 2008*; *Tóth et al., 2018*; *de la Prida and Huberfeld, 2019*) during associative memory processing (*Kucewicz et al., 2014*) and memory retrieval (*Vaz et al., 2019*). In addition, pvBC self-inhibition may contribute to pyramidal cell disinhibition during the induction of L2/3 long-term potentiation associated with learning (*Williams and Holtmaat, 2019*).

# Materials and methods

## Ethics statement

All procedures were performed according to the Declaration of Helsinki with the approval of the University of Szeged Ethical Committee and Regional Human Investigation Review Board (ref. 75/2014). For all human tissue material, written consent was obtained from patients prior to surgery. Tissue

obtained from underage patients was provided with agreement from a parent or guardian. In 5 of the 20 pvBC-PC cell pairs reporting IPSC conductance, some other data parameters (excluding conductance reported here) have been reported in a previous manuscript (*Szegedi et al., 2017*).

## Human brain slices

Neocortical slices were sectioned from material that had to be removed to gain access for the surgical treatment of deep-brain targets from the left and right frontal, temporal or occipital areas. In some cases (aneurysm, hydrocephalus, cortical metaplasia when removed tissue was not in the pathological focus) tissue from neocortical operations was used. The patients were 10–85 years of age, and samples from males and females from either the left or right hemisphere were included. Details including patient gender, age, resected neocortical area and pathological target diagnosis are reported for all tissue samples used in this study in *Supplementary file 1*. Anesthesia was induced with intravenous midazolam and fentanyl (0.03 mg/kg, 1–2 lg/kg, respectively). A bolus dose of propofol (1–2 mg/kg) was administered intravenously. The patients received 0.5 mg/kg rocuronium to facilitate endotracheal intubation. The trachea was intubated, and the patient was ventilated with an $O_2/N_2O$ mixture at a ratio of 1:2. Anesthesia was maintained with sevoflurane at a care volume of 1.2–1.5. Following surgical removal, the resected tissue blocks were immediately immersed into a glass container filled with ice-cold solution in the operating theatre. The solution contained (in mM): 130 NaCl, 3.5 KCl, 1 $NaH_2PO_4$, 24 $NaHCO_3$, 1 $CaCl_2$, 3 $MgSO_4$, 10 D(+)-glucose and was saturated with 95% $O_2$/5% $CO_2$. The container was placed on ice in a thermally isolated transportation box where the liquid was continuously gassed with 95% $O_2$/5% $CO_2$. Then, the tissue was transported from the operating theatre to the electrophysiology lab (door-to-door in maximum 20 min), where slices of 350 µm thickness were immediately prepared from the block with a vibrating blade microtome (Microm HM 650 V). The slices were incubated at 22–24°C for 1 hr, when the slicing solution was gradually replaced by a pump (6 ml/min) with the solution used for storage (180 ml). The storage solution was identical to the slicing solution, except for 3 mM $CaCl_2$ and 1.5 mM $MgSO_4$.

## Pv+ cells in mouse brain slices

Transversal slices (350 µm) from somatosensory cortex were prepared (*Kotzadimitriou et al., 2018*) from 4- to 6-week-old heterozygous male CB6-Tg(Gad1-EGFP)G42Zjh/J -mice (The Jackson Laboratory, stock 007677, GAD67-GFP G42 line) expressing td-Tomato fluorophore preferably in parvalbumin GABAergic neurons (*Chattopadhyaya et al., 2004*). Cells were confirmed to be fast-spiking showing fast spike kinetics and a high-frequency non-accommodation firing pattern for suprathreshold depolarizing 500 ms pulses. Cells were visualized with streptavidin Alexa488 (1:2000, Jackson ImmunoResearch Lab, Inc) and analyzed by eye under epifluorescence microscopy to exclude axoaxonic cells. Three cells were selected for pv immunoreactivity, and they were all immunopositive for pv (see *Supplementary file 1*).

## Electrophysiology

Recordings were performed in a submerged chamber (perfused 8 ml/min) at 36–37°C. Cells were patched using a water-immersion 20x objective with additional zoom (up to 4x) and infrared differential interference contrast video microscopy. Micropipettes (5–8 MΩ) for whole-cell patch-clamp recording were filled with intracellular solution with physiological or elevated intracellular chloride $[Cl^-]_i$. The content of the solution for voltage clamp recordings with physiological $[Cl^-]_i$ was (in mM): 126 K-gluconate, 8 NaCl, 4 ATP-Mg, 0.3 $Na_2$–GTP, 10 HEPES, and 10 phosphocreatine (pH 7.0–7.2; 300 mOsm) with 0.3% (w/v) biocytin. Current clamp recordings with elevated $[Cl^-]_i$ contained 130 mM KCl instead. Recordings were performed with a Multiclamp 700B amplifier (Axon Instruments) and low-pass filtered at 6–8 kHz cut-off frequency (Bessel filter). Series resistance (Rs) and pipette capacitance were compensated in current clamp mode and pipette capacitance in voltage clamp mode. Rs was monitored and recorded continuously during the experiments. Voltage clamp recordings were discarded if the Rs was higher than 25 MΩ or changed by more than 20%. Liquid junction potential error was corrected in all membrane potential values. The access resistance of the recording electrode was measured, and its effect on the clamping potential error was corrected in nominal somatic potential reading. Resting membrane potential (Em) was recorded 1–3 min after break-in to whole cell. Em of human pvBCs (−74.15 ± 0.99 mV, n = 64) was not different from Em of mouse

pvBCs (−77.21 ± 1.46 mV, n = 19) (p=0.537, Shapiro-Wilk normality test; p=0.130, Student's t-test). Em in nonFSINs was −71.82 ± 1.97 mV (p=0.313, Shapiro-Wilk normality test, n = 22). Cell capacitance and input resistance were measured in current clamp using −50–100 pA, 600 ms steps delivered at resting membrane potential.

Single-spike firing in current clamp or in voltage clamp was induced by a 50 ms depolarizing suprathreshold step from the resting membrane potential. The autaptic GBZ-sensitive outward current amplitude was 150.2 pA, 60.0 to 177.5 pA (median, quartiles, n = 14). Synaptic IPSCs in pvBC-PC pairs were recorded at steady postsynaptic −43 mV to −55 mV clamping potential. Conductance was calculated from averaged (at least 12 events) current peak amplitude and the $GABA_A$ current driving force for each cell by Ohm's law formula. The $GABA_A$ current reversal potential with the recording solution containing physiological Cl⁻ was determined as −73 mV by Hodgkin–Katz voltage equation using 0.1 relative permeability of bicarbonate to Cl⁻ with 7.4. extra- and 7.0 intracellular pH. Similarly, reversal potential for AHP (−95 mV) was calculated from transmembrane K+ concentration gradient. Evoked autaptic and synaptic responses were analyzed using Clampfit or Spike2 programs as described in *Szegedi et al. (2017)*. Cell input resistance and capacitance were measured in the current clamp at the resting membrane potential.

## Dynamic clamp

To emulate the EPSPs and IPSCs in basket cells and other interneurons, a software-based dynamic clamp system was employed. Current injections were calculated by the StdpC2017 software (*Kemenes et al., 2011*) through a MIO-16E-4 analog/digital card (National Instruments Inc, Hungary) based on voltage signals of the electrode. We ran the dynamic clamp on a separate computer from our experimental data acquisition system (recording cell membrane potential) to record and verify dynamic clamp output (conductance and EPSCs). Sub- and suprathreshold EPSCs were evoked using a decay time constant of 1.25 ms and a reversal potential of 0 mV. The peak conductance (1.5–8 nS) for subthreshold EPSP was set to evoke a 2–9 mV peak amplitude response, and the onset of post-spike EPSC was triggered by a preceding spike being time locked to it with a 5 ms onset delay. Depolarizing step (square pulse 0.5 ms) triggering the spike had peak conductance up to 20 nS. Autaptic IPSCs had 1–10 nS conductance, a decay time constant of 5 ms and a reversal potential of −78 mV. The onset of IPSC was triggered by a preceding spike (adjusting a threshold to trigger IPSC at −20 mV) with a 1 ms delay.

## Single cell model

In the simulation of somatic EPSP inhibition by autaptic shunting, real experimental data from individual basket cells were used. Simulation of a basket cell membrane potential was performed using a NEURON 7.6.5 simulator (*Carnevale and Hines, 2006*). The membrane capacitance was 1 μF/cm$^2$, and the size of the soma was determined so that the total cell capacitance matched the actual measured value in each cell. $G_{leak}$ was retrieved from experimental data, and $G_{leak}$ reversal potential was set to −68 mV (to establish −68 mV as the resting membrane potential). EPSPs were modeled using rise tau 0.2 ms, decay tau 1.2 ms, conductance of 10 nS with reversal potential 0 mV. The EPSP onset delay was set to 3–20 ms from '0' time point (the measured time point of the action potential inward current peak) with varying steps (1 ms interval).

## Data analysis

Data were acquired with Clampex software (Axon Instruments) and digitized at 10–50 kHz. The data were analyzed off-line with pClamp (Axon Instruments), Spike2 (version 8.1, Cambridge Electronic Design), OriginPro (OriginLab Corporation) and IgorPro (WaveMetrics Inc) and SigmaPlot14 software. Spike kinetics (action current width) and synaptic parameters were analyzed as described previously (*Szegedi et al., 2016*; *Szegedi et al., 2017*). Autaptic currents and potentials were defined by subtracting trace averages (of at least six events) in gabazine from traces evoked by preceding trials. Peak conductance was calculated from currents in voltage clamp according to Ohm's law. The delay-to-peak value was defined from the action potential inward current peak ('0' time point) to maximal (AHP or autaptic) outward current value.

## Statistics

Data are presented as the mean ± s.e.m, when showing n $\geq$ 7 with a parametric distribution. Normality was tested with the Shapiro-Wilk test P value > 0.05. Otherwise, the data are shown as the median with interquartile range (of lower and upper quartile) unless stated otherwise. Correspondingly, for statistical analysis, t-test, Mann-Whitney U -test, Chi-square test, Wilcoxon signed-rank test or ANOVA on ranks (with Dunn's *post hoc* test), was used. Correlations were tested using Pearson's or Spearman's correlation, respectively. In addition, two sample Kolmogorov–Smirnov test was used to test nonparametric probability distributions. Differences were accepted as significant if p<0.05.

## Tissue fixation and cell visualization

Biocytin-filled cells were visualized with either Alexa488- (1:500) or Cy3-streptavidin (1:400, Jackson ImmunoResearch Lab, Inc) for anatomical and immunohistochemical investigation. After electrophysiological recording, slices were immediately fixed in a fixative containing 4% paraformaldehyde and 15% picric acid in 0.1 M phosphate buffer (PB, pH = 7.4) at 4°C for at least 12 hr and then stored at 4°C in 0.1 M PB with 0.05% sodium azide as a preservative. For some slices, 0.05% glutaraldehyde was added in fixative for electron microscopy studies. All slices were embedded in 10% gelatin and further sectioned into slices of 50 μm thickness in the cold PB using a vibratome VT1000S (Leica Microsystems). After sectioning, the slices were rinsed in 0.1 M PB (3 × 10 min) and cryoprotected in 10–20% sucrose solution in 0.1 M PB. After this, the slices were frozen in liquid nitrogen and thawed in 0.1 M PB. Finally, they were incubated in fluorophore-conjugated streptavidin (1:400, Jackson ImmunoResearch Lab, Inc) in 0.1 M Tris-buffered saline (TBS, pH 7.4) for 2.5 hr (at 22–24°C). After washing with 0.1 M PB (3 × 10 min), the sections were covered in Vectashield mounting medium (Vector Laboratories Inc), placed under cover slips, and examined under an epifluorescence microscope (Leica DM 5000 B).

## Cell reconstruction and anatomical analyses

Sections selected for immunohistochemistry and cell reconstruction were dismounted and processed (see 'Immunohistochemistry' paragraph). Some sections for cell structure illustrations were further incubated in a solution of conjugated avidin-biotin horseradish peroxidase (ABC; 1:300; Vector Labs) in Tris-buffered saline (TBS, pH = 7.4) at 4°C overnight. The enzyme reaction was revealed by the glucose oxidase-DAB-nickel method using 3'3-diaminobenzidine tetrahydrochloride (0.05%) as the chromogen and 0.01% $H_2O_2$ as the oxidant. Sections were further treated with 1% $OsO_4$ in 0.1 M PB. After several washes in distilled water, sections were stained in 1% uranyl acetate and dehydrated in an ascending series of ethanol concentrations. Sections were infiltrated with epoxy resin (Durcupan) overnight and embedded on glass slides. For the cells visualized in the figures, three-dimensional light microscopic reconstructions from one or two sections were carried out using the Neurolucida system with 100x objective (Olympus BX51, Olympus UPlanFl). Images were collapsed in the z-axis for illustration. The somatodendritic region in the 50 μm-thick section was studied for close appositions with filled axons traced back to the soma. Neurolucida explorer software was used to measure the distance of close appositions in dendrites to the soma in images visualized using the same computer software.

## Immunohistochemistry

Free-floating sections were washed three times in TBS-TX 0.3% (15 min) at 22–24°C and then moved to 20% blocking solution with horse serum in TBS-TX, 0.3% for parvalbumin (pv) staining and 10% blocking solution for vesicular GABA transporter (vGAT) staining. For sections from tissue fixed with glutaraldehyde-containing solution, treatment with pepsin was applied to improve immunohistochemical staining (*Gulyás et al., 2010*). Similar treatment was applied prior to immunohistochemical reaction for ankyrin-G protein located in axon initial segment (see *Figure 1—figure supplement 1c*). The sections were treated with 1 mg/ml pepsin (catalog #S3002; Dako) in 0.2 M HCl with 0.1 M PB at 37°C for 6 min and then washed in 0.1 M PB. All sections were incubated in primary antibodies diluted in 1% serum in TBS-TX 0.3% over three nights at 4°C, and placed in relevant fluorochrome-conjugated secondary antibodies in 1% blocking serum in TBS-TX 0.3% overnight at 4°C. Sections were first washed in TBS-TX 0.3% (3 × 20 min) and later in 0.1 M PB (3 × 20 min) and mounted on glass slides with Vectashield mounting medium (Vector Lab, Inc).

The characterizations of antibodies used in humans; pv = (mouse anti-pv, 1:500, Swant, Switzerland, www.swant.com, clone: 235). vGAT = (rabbit anti-vgat, 1:500, Synaptic Systems, Germany, www.sysy.com, AB_887871). AnkyrinG = (mouse anti-ankG, 1:100, Santa Cr. B., sc-12719). Fluorophore-labeled secondary antibodies were (DAM DyLight 488 donkey anti mouse, 1:400, Jackson ImmunoResearch Lab. Inc, www.jacksonimmuno.com) or (DAM Cy3 donkey anti mouse, 1:400, Jackson ImmunoResearch Lab. Inc, www.jacksonimmuno.com) and (DARb Cy5 donkey anti-rabbit, 1:500, Jackson ImmunoResearch Lab. Inc). Antibodies in mice were (goat anti-pv, 1:2000, Swant, AB_10000343) and (DAGt Cy5 donkey anti-goat, 1:400, Jackson ImmunoResearch Lab. Inc). The labeling of neurons by biocytin and immunoreactions was evaluated using first epifluorescence (Leica DM 5000 B) and then laser scanning confocal microscopy (Zeiss LSM880). Both pv and vGAT immunoreactions were studied in axon. All micrographs presented are confocal fluorescence images.

## Electron microscopy

Sections containing pvBC soma were re-embedded, and 65 nm thick ultrathin sections were cut with an ultramicrotome (RMC MT-XL). Ribbons of the sections were collected on Formvar-coated copper grids and examined with a JEOL JEM-1400Plus electron microscope. Images were taken by an 8 M pixel CCD camera (JEOL Ruby).

## Acknowledgements

This work was supported by the Hungarian Academy of Sciences and Eötvös Loránd Research Network (GT), the National Research, Development and Innovation Office of Hungary GINOP-2.3.2-15-2016-00018 (GT), OTKA K128863 (GM, GT, KL), the Ministry of Human Capacities Hungary grant 20391-3/2018/FEKUSTRAT (GT, KL), the ERC INTERIMPACT project (GT), the National Brain Research Program Hungary (KL, MP, VS, JB, GM and GT) and by University of Szeged Open Access Fund (Grant number: 4373). We acknowledge Ms Leona Mezei and Ms Emöke Bakos for technical assistance.

## Additional information

### Funding

| Funder | Grant reference number | Author |
|---|---|---|
| National Research Development and Innovation Office | National Brain Research Programme | Viktor Szegedi<br>Melinda Paizs<br>Karri Lamsa |
| ERC | INTERIMPACT | Gabor Tamas |
| Hungarian Academy of Sciences | | Viktor Szegedi<br>Gábor Molnár |
| University of Szeged | Open Access Fund 4373 | Viktor Szegedi<br>Karri Lamsa |
| Eotvos Lorand Research Network | | Gabor Tamas |
| National Research Development and Innovation Office | GINOP-2.3.2-15-2016-00018 | Gabor Tamas |
| National Research Development and Innovation Office | OTKA K128863 | Gábor Molnár<br>Gabor Tamas<br>Karri Lamsa |
| Ministry of Human Capacities | 20391-3/2018/FEKUSTRAT | Gabor Tamas<br>Karri Lamsa |

The funders had no role in study design, data collection and interpretation, or the decision to submit the work for publication.

## Author contributions
Viktor Szegedi, Conceptualization, Formal analysis, Validation, Investigation, Visualization; Melinda Paizs, Judith Baka, Investigation, Visualization; Pál Barzó, Gábor Molnár, Resources; Gabor Tamas, Resources, Funding acquisition; Karri Lamsa, Conceptualization, Formal analysis, Supervision, Funding acquisition, Validation, Investigation, Visualization

## Author ORCIDs
Viktor Szegedi  https://orcid.org/0000-0003-4191-379X
Gabor Tamas  http://orcid.org/0000-0002-7905-6001
Karri Lamsa  https://orcid.org/0000-0002-4609-1337

## Ethics
Human subjects: All procedures were performed according to the Declaration of Helsinki with the approval of the University of Szeged Ethical Committee and Regional Human Investigation Review Board (ref. 75/2014). For all human tissue material, written consent was obtained from patients prior to surgery. Tissue obtained from underage patients was provided with agreement from a parent or guardian.

Animal experimentation: All procedures were performed with the approval of the University of Szeged (no. I-74-8/2016) and in accordance with the Guide for the Care and Use of Laboratory Animals (2011; http://grants.nih.gov/grants/olaw/guide-for-the-care-and-use-oflaboratory-animals.pdf ).

## Decision letter and Author response
Decision letter https://doi.org/10.7554/eLife.51691.sa1
Author response https://doi.org/10.7554/eLife.51691.sa2

## Additional files

### Supplementary files
• Supplementary file 1. Details of recorded human neurons and information on the resected human cortical tissue. Columns from left to right show the figure in which the cell recording data are shown (cells showing no evidence for autapse are indicated as 'no aut'), cell filing code, indication for successful recovery of cell with streptavidin visualization (strept) (NA indicates unsuccessful cell recovery), immunoreaction (positive as + and negative as -) for parvalbumin (pv) and vesicular GABA transported (vGAT), action potential inward current width (acw), firing frequency accommodation shown in cells where it was tested (first number shows firing frequency as 'Hz' during first 100 ms of a robust although not always maximal depolarizing pulse. Second number shows firing frequency accommodation 'acc.' = firing frequency at 400–500 ms during the depolarization divided by firing frequency during first 100 ms), and resting membrane potential ($E_m$). Details of resected tissue are shown in blue, showing patient gender, age, hemisphere, cortical area and diagnosed primary pathology. There is 'no info' about exact neocortical area in some ventriculostomy operations.

• Supplementary file 2. Recorded parameters used in computational simulation of human pvBCs.

• Transparent reporting form

### Data availability
All data generated or analyzed during this study are included in the manuscript and supporting files.

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
