## [Decision Letter]

**Acceptance summary:**

In a technical tour de force, the authors obtained high quality intracellular recordings from interneurons of resected human neocortex, and integrated the findings with elegant immunohistochemistry and electron microscopy. Through these approaches, this paper proves that autaptic inhibition in PV basket cells is a robust feature of the human superficial neocortex. Further they show that this form of autaptic self-inhibition functionally modulates spiking by affecting the probability and timing of subsequent spikes.

**Decision letter after peer review:**

Thank you for submitting your article "Robust perisomatic GABAergic self-innervation inhibits basket cells in the human and mouse supragranular neocortex" for consideration by *eLife*. Your article has been reviewed by two peer reviewers, and the evaluation has been overseen by a Reviewing Editor and Gary Westbrook as the Senior Editor. The following individual involved in review of your submission has agreed to reveal their identity: Alberto Bacci (Reviewer #2).

The reviewers have discussed the reviews with one another and the Reviewing Editor has drafted this decision to help you prepare a revised submission.

Summary:

Szegedi and colleagues describe the presence of self-inhibitory autaptic connections in supragranular layers of the human neocortex. The authors used healthy live resected neocortical tissue obtained from patients requiring deep brain surgery. Previously, the existence of autaptic contacts in neocortical interneurons was anatomically demonstrated in cats, and described to be functional in fast-spiking interneurons of rodents and humans. Despite these previous demonstrations that autaptic transmission was strong and common in PV cells of deep neocortical layers it is unknown whether it is similarly pervasive with equivalent functional effects on excitability in superficial neocortex, especially in humans.

This manuscript reports data of very high quality, as a result of a technical tour de force: the authors recorded from many interneurons of resected human neocortex, performed immunohistochemistry, electron microscopy, multiple patch clamp recordings and dynamic clamp. By combining electron microscopy and electrophysiology, this paper elegantly demonstrates the existence of functional autaptic inhibition in PV basket cells of the human superficial neocortex. Further they show that autaptic self-inhibition functionally modulates spikes by affecting the probability and timing of subsequent spikes.

Essential revisions:

1) Novelty statements. The authors should be more generous in their comments about previous work. For example, the statement "only a single study has reportedly investigated [autapses] in the human neocortex" (Introduction) sounds like the authors don't believe this work. I suggest deleting "reportedly". The main novelty of this paper is the use of human tissue but, as the authors mention, a paper published in 2012 (Jiang et al.) has already studied human cortical autapses to a somewhat more limited extent than the results presented in the current manuscript. The next paragraph begins "Few notable studies in rodents have demonstrated…". Again, this implies that there have mostly been non-notable studies, which is inaccurate and unkind – there have been plenty of very good studies of autapses in the past. We suggest replacing this with, "A number of studies in rodents have demonstrated…". A third example is at the end of the third paragraph of the Introduction: "… but very little is known about autapses themselves… ". It would be better to replace "themselves" with "in human brain", to avoid annoying the authors of the many papers on autapses in rodents. There are similar statements in the Discussion that should be modified.

2) Identification of pvBCs. The classification criteria should be made clearer. According to Table 1, the immunohistochemistry was often inconclusive or not done. The legend mentions maximal firing frequency but this does not appear in the table. The "acw" (action current width) measure is non-standard and the choice of an appropriate cutoff between FS and non-FS cells is not discussed. At the very least, there should be clearer evidence that "acw" is a good proxy for AP width and is strongly diagnostic for FS cells, and FS cells alone. It would have been better to record an action potential family using increasing current step sizes for each cell, allowing better characterization of the cell identity, and I would be surprised if this was not done routinely.

3) Correlation between AHP and membrane leak conductances. This is specifically discussed at the end of the Discussion, yet the raw data are not shown in the paper; instead, only the mean data are briefly mentioned (p 7). It would be helpful to show the raw data as a supplementary figure, given that it appears to be a key discussion point.

4) The presentation of data in Figure 3 are not straightforward, and the inclusion criteria for neurons in this analysis is not clear. Why did they use 10 cells showing the strongest autaptic conductance? What do they mean with 'strongest'? In addition, in Figure 3A, why did the authors show averages without showing an estimate of statistical variability (e.g. SEM or SD)? Traces in Figure 3B are obscure. One might infer that the hyperpolarizing portion of the trace is the AHP, however no spike is shown. Did the modeled neuron generate a spike? Wouldn't the AHP kinetics depend on the spike waveform? In that case, wouldn't the actual spike generate a shunting component, which might also affect membrane excitability during the AHP and autaptic transmission?

5) The authors write: "In all simulations, the GABA_A_ reversal potential was set at the membrane potential of EPSP onset (Supplementary file 2)." I could not understand this sentence. Did the authors change E_GABA-A_ depending on when the EPSP was induced? In any case, Supplementary file 2 does not report anything related to reversal potential.

6) Figure 4: why are conductance (grey) traces shown as negative waveforms? The same is shown in Figure 5Ai.

7) In the Materials and methods section, the authors write that they elicit spikes using depolarizing steps of 50 ms. If this is not an typo 50 ms would be too long to isolate autaptic responses. On the same point, there is insufficient presentation of the actual protocol implemented to induce autaptic responses in FS interneurons while minimizing current distortions induced by capacitive transients and the large unclamped action currents. I did not understand what 'Vcl=43 mV' means in panel Aii of Figure 2—figure supplement 1. I assume this is not the reversal potential for GABA-mediated responses, but rather the holding potential.

8) In the Materials and methods section it is written: 'Single-spike firing in AACs was elicited with 1-2 ms suprathreshold depolarization.' Is this a typo? I assume the authors did not analyze axo-axonic cells.

9) The authors should mention in their Discussion the possibility that the incidence of autaptic connectivity, as well as strength and biophysical properties of autaptic and synaptic responses could be influenced by resected axons while preparing acute slices. The authors should provide some indications on the actual health of their slices. Is autaptic incidence and strength is affected by the age of the patient?

---

## [Author Response]

Essential revisions:1) Novelty statements. The authors should be more generous in their comments about previous work. For example, the statement "only a single study has reportedly investigated [autapses] in the human neocortex" (Introduction) sounds like the authors don't believe this work. I suggest deleting "reportedly". The main novelty of this paper is the use of human tissue but, as the authors mention, a paper published in 2012 (Jiang et al.) has already studied human cortical autapses to a somewhat more limited extent than the results presented in the current manuscript.

This sentence has now been changed.

The next paragraph begins "Few notable studies in rodents have demonstrated…". Again, this implies that there have mostly been non-notable studies, which is inaccurate and unkind – there have been plenty of very good studies of autapses in the past. We suggest replacing this with, "A number of studies in rodents have demonstrated…".

We thank for this comment and the sentence has been reformulated as suggested.

A third example is at the end of the third paragraph of the Introduction: "… but very little is known about autapses themselves… ". It would be better to replace "themselves" with "in human brain", to avoid annoying the authors of the many papers on autapses in rodents. There are similar statements in the Discussion that should be modified.

We have changed this sentence as suggested. In addition, we have modified sentences in Discussion as follows:

– “Therefore, it is unclear whether robust autaptic self-inhibition is a general feature in the neocortex operating in various neocortical layers and different species (including human) or whether GABAergic self-inhibition is a specialization in deep neocortical interneurons.”

– “In addition, our results here with human and mouse cells (together with earlier studies on rat neurons) confirm that autapses have similar occurrence and strength in pvBCs in rodent and human brain slices.”

– “GABAergic self-inhibition conductance in human pvBCs is comparable to the synaptic inhibition these interneurons exert on neighboring layer 2/3 neurons.”

2) Identification of pvBCs. The classification criteria should be made clearer. According to Table 1, the immunohistochemistry was often inconclusive or not done. The legend mentions maximal firing frequency but this does not appear in the table. The "acw" (action current width) measure is non-standard and the choice of an appropriate cutoff between FS and non-FS cells is not discussed. At the very least, there should be clearer evidence that "acw" is a good proxy for AP width and is strongly diagnostic for FS cells, and FS cells alone. It would have been better to record an action potential family using increasing current step sizes for each cell, allowing better characterization of the cell identity, and I would be surprised if this was not done routinely.

We agree that cell identification criteria were not explicitly presented in the original version of the manuscript and we apologise for it. We have now made cell identification results and criteria clearer to a reader. Therefore, we have added a new supplementary figure, Figure 1—figure supplement 1. We have performed more immunohistochemical analyses for the cells, and we also include firing frequency accommodation information for cells where it was tested. Thus, our update to clarify cell type identification comprises:

a) New Figure 1—figure supplement 1 clarifies cell type identification and explains how it is based on the anatomical analysis of recorded neuron’s visualised axon and the cell’s action potential inward current kinetics. Basket cells form basket-like structures around L2/3 cell somata. New Figure 1—figure supplement 1 illustrates this. Thirty-nine of 46 fast-spiking cells tested specifically for autapses were successfully visualised and anatomically confirmed as basket cells. However, because seven fast-spiking cells in the data set remain unsuccessfully visualized (and we do not have anatomical data from these cells to confirm their cell type identity), we have reformulated the first paragraph in Results section as follows: “39 basket cells were identified by their axon forming boutons around unlabeled L2/3 neurons (Figure 1—figure supplement 1) (Szegedi et al., 2017). Seven unsuccessfully visualized fast-spiking interneurons were included as putative pvBCs (Figure 1—figure supplement 1).” In addition, 21 of 22 nonFSINs (showing slow axon potential inward current component) were successfully visualised and anatomically identified as non-basket cell interneurons (see also Szegedi et al., 2016; 2017).

b) Histograms in the new Figure 1—figure supplement 1 show the action potential inward current width (acw) separately for all identified basket cells and putative basket cells (including those studied for synaptic transmission) as well as in nonFSINs of this study. The “acw”-value of basket cells and nonFSINs does not overlap (P = 0.0001, two-sample Kolmogorov-Smirnov test). In addition, the unsuccessfully visualized fast-spiking cells show acw-value similar to pvBC. In Figure 1—figure supplement 1 we suggest it is more likely that these neurons are basket cells than axo-axonic cells (AACs), since our supplementary data indicate (in line with an earlier animal study) that AACs do not have autapses (see Figure 1—figure supplement 1).

c) We have performed new immunohistochemical tests and analyses for all cells which in the original manuscript version were reported as “pv non-conclusive” or “pv not tested”. As the updated Supplementary file 1 shows now, the pv immuno re-testing concerns 16 fast-spiking basket cells in the actual data set (studied for autapses) and 9 fast-spiking basket cells in the supplementary material (cells studied for synapses).

Importantly, these slices were initially fixed using glutaraldehyde-containing fixative and unfortunately, this compromises immunoreactions. As we report in Materials and methods, glutaraldehyde (gluAH) in fixation solution is required for high quality electron microscopy (EM) investigation. In the revised manuscript, we have used brief protein degradation pre-treatment with pepsin in these slices. The protocol has been used earlier to improve immunoreactions in gluAH-fixed rodent brain slices (we have clarified this point now in Materials and methods with a reference to Gulyás et al., 2010). The protocol helped us to confirm 11 of the 25 cells as pv+. Yet, sections for 14 pepsin pre-treated cells still failed to generate successful pv immunoreaction although we had two attempts using pepsin treatment. Those cells still remain “pv non conclusive” in the updated Supplementary file 1.

d) We have now visualised 21 of 22 non-FSIN of this study to anatomically analyse their axon and somatodendritic structure (visualization of one nonFSIN was unsuccessful). In addition, we have systematically applied immunoreaction for pv in these cells although many of these slices were fixed with gluAH-containing solution. Only pv- cells were found in the nonFSINs. The results are now shown in the updated Supplementary file 1.

e) We have also added action potential firing pattern analysis in all cells where firing response to a depolarizing step (500 ms) was taken at the end of the experiment (please see Supplementary file 1). The firing frequency data does not systematically show the ultimate maximal firing frequency of the cell. However, it demonstrates high-frequency firing capacity and small firing frequency accommodation of cells with narrow “acw”. This is important because it clearly discriminates our cells from another human interneuron cell type with a relatively fast spike waveform: a rosehip cell. Yet, unlike pv+ cells the rosehip interneurons show low firing frequency and stuttering spiking pattern (Boldog et al., 2018, cited in the manuscript).

3) Correlation between AHP and membrane leak conductances. This is specifically discussed at the end of the Discussion, yet the raw data are not shown in the paper; instead, only the mean data are briefly mentioned (p 7). It would be helpful to show the raw data as a supplementary figure, given that it appears to be a key discussion point.

Figure 2—figure supplement 2 illustrates this.

4) The presentation of data in Figure 3 are not straightforward, and the inclusion criteria for neurons in this analysis is not clear. Why did they use 10 cells showing the strongest autaptic conductance? What do they mean with 'strongest'? In addition, in Figure 3A, why did the authors show averages without showing an estimate of statistical variability (e.g. SEM or SD)?

We have now restructured Figure 3 as well as the corresponding Results section in the manuscript text. New analyses and figures include 13 of 14 human pvBCs as well as all 11 mouse pvBCs. Both analyses show mean ± sem or median and quartiles (depending on whether the peak amplitude and the decay time constant data show parametric distribution or not). We have excluded one human pvBC in the G_aut_ kinetics plot and simulation experiment because its small autaptic current does not allow a confident setting of decay tau. We have explained this in Results.

Traces in Figure 3B are obscure. One might infer that the hyperpolarizing portion of the trace is the AHP, however no spike is shown. Did the modeled neuron generate a spike?

Spike outward current (i.e. K^+^ current component responsible for the action potential afterhyperpolarization) but not the inward current component is simulated. Reason for this is that the inward current, whose generation is responsible for both G_ahp_ and G_aut_ with specific delay, precedes these conductances, and our interest in this study is in post-spike time window at time points where a second spike could be generated in the cell (as we later on demonstrate in Figure 5). In human fast-spiking pvBCs this means an inter-spike interval of 3 ms or longer as can be seen in data now presented in the Supplementary file 1 ‘firing accommodation’ column (some human pvBCs fire at 370 Hz and many at 300 Hz).

In the simulation experiments the spike inward current timing is in a central role as we now clarify in the Results; its measured peak establishes “0-time point” for G_ahp_ and G_aut_ conductance values. The measured delay to the action potential inward current peak is used for G_ahp_ and G_aut_ onset and peak times. We report these values in the manuscript Results section and in the Supplementary file 2. In addition, we have now indicated the spike inward current peak as “0-time point” in traces showing individual basket cell membrane potential in the simulation experiments.

Wouldn't the AHP kinetics depend on the spike waveform? In that case, wouldn't the actual spike generate a shunting component, which might also affect membrane excitability during the AHP and autaptic transmission?

We agree that action current outward component (net outward current) onset and rise phase can include some Na^+^ ‘inward current’ conductance. Yet, the AHP outward current peak and any time point after it (particularly post-spike time points with at least 3 ms delay from action potential inward current peak we focus on) are unlikely affected.

5) The authors write: "In all simulations, the GABA_A_ reversal potential was set at the membrane potential of EPSP onset (Supplementary file 2)." I could not understand this sentence. Did the authors change E_GABA-A_ depending on when the EPSP was induced? In any case, Supplementary file 2 does not report anything related to reversal potential.

We agree this was presented vaguely in the original manuscript version and we apologize for it. We have rewritten the entire paragraph on the simulation experiments. We explain now more clearly the details how GABA_A_ reversal potential and other parameters were set. In addition, we have updated the Supplementary file 2 with E_GABA-A_ values.

6) Figure 4: why are conductance (grey) traces shown as negative waveforms? The same is shown in Figure 5Ai.

This has now been corrected both in Figure 4 and 5.

7) In the Materials and methods section, the authors write that they elicit spikes using depolarizing steps of 50 ms. If this is not an typo 50 ms would be too long to isolate autaptic responses.

We agree that 50 ms depolarizing step is longer than what is required to study a time-locked autaptic response by spike. Yet, we see that holding a cell at specific depolarizing potential for 50 ms is not harmful either for autapse recording. As a matter of fact, a 25 ms or longer step helps when we determine autaptic current or I_ahp_ decay time constant. Yet, reason why we have used such a long step here is that we wanted to regularly monitor possible network activity up to 50 ms after a FSIN spike; previous studies have shown that the unitary spike in some human L2/3 interneurons can initiate local network activity time-locked to interneuron spike (Molnar et al., 2008; Komlosi et al., 2012, both cited in the manuscript). To monitor this we used 50 ms depolarizing step so that in case such activity is generated, the voltage clamp step recovery wouldn’t contaminate synaptic currents possibly occurring during the (up to 50 ms long) activity. However, we did not detect such ‘complex event’ activity evoked by any of our pvBCs or nonFSINs, and because we think that the finding is not relevant here we haven’t reported it and therefore also specific reasoning for the depolarizing step being 50 ms was missing.

On the same point, there is insufficient presentation of the actual protocol implemented to induce autaptic responses in FS interneurons while minimizing current distortions induced by capacitive transients and the large unclamped action currents.

We have now added in Materials and methods the following sentences to clarify the autapse recording protocol: “We set the depolarizing current moderately stronger than rheobase current aiming to initiate single action potential or two action potentials with long inter-spike interval. Autaptic current or potential was always studied for the first action potential generated by the step.” When it comes to autapse current distortion by a voltage clamping step due to the cell fast capacitance charging, or by the action potential escape current inward component, we clarify it now in the Materials and methods that “spike-evoked autaptic outward current was uncovered by subtracting response to the depolarizing step in gabazine (average of 6 after wash-in of GBZ, 10 μM) from baseline responses.The subtraction of the responses reveals gabazine-sensitive autaptic current reliably, since the cell ‘fast capacitance’ charging current, as well as the action potential escape inward current remain stable as far as the resistance between pipette and the cell soma stays relatively unchanged. For this reason the voltage clamp experiments recording the baseline and GBZ wash-in are fairly short in this study (because during long recordings the access resistance is more likely to change).”

Finally, although we cannot claim that a single voltage clamp recording system could fully encage full kinetic feature of the autaptic and the AHP conductance rise phase or peak conductance, we can confirm that during the voltage clamp recordings the cell membrane potential did not change indicating that the gain in voltage clamp system has been sufficient to clamp the somatic membrane potential during the events. Therefore we can argue that G_ahp_ and G_aut_ values calculated from the current components represent fairly realistic conductance values in the soma.

I did not understand what 'Vcl=43 mV' means in panel Aii of Figure 2—figure supplement 1. I assume this is not the reversal potential for GABA-mediated responses, but rather the holding potential.

We apologize this was not clarified well in the original manuscript version. We now mention in the figure legend that “Vcl = -43 mV” means that the holding potential in voltage clamp is -43 mV for the traces.

8) In the Materials and methods section it is written: 'Single-spike firing in AACs was elicited with 1-2 ms suprathreshold depolarization.' Is this a typo? I assume the authors did not analyze axo-axonic cells.

We apologize the sentence is irrelevant and it has been removed.

9) The authors should mention in their Discussion the possibility that the incidence of autaptic connectivity, as well as strength and biophysical properties of autaptic and synaptic responses could be influenced by resected axons while preparing acute slices.

We have added this point in the Discussion as follows: “Information on the self-innervation subcellular target motifs as well as knowledge on autaptic inhibition occurrence and strength in identified neurons is essential for understanding the operation of autaptic microcircuits in individual neurons and in a neuronal network. Our results in human neocortex shed more light on this, but it is worth keeping in mind that the incidence of autaptic connectivity, and hence the strength of autaptic inhibition particularly in dendrites can be influenced by resected axons of brain slices”.

The authors should provide some indications on the actual health of their slices.

We now report the resting membrane potential (E_m_) for all neurons as an indicator of health of cells (and slices). The values are not different between human and mouse pvBCs (P = 0.146). We report this now in the manuscript. In addition, we show E_m_ value for every cell in the updated Supplementary file 1.

Is autaptic incidence and strength is affected by the age of the patient?

We have performed these relevant analyses. As data plots in Author response image 1 demonstrate our data shows (left) similar patient age for pvBCs with autapse (red, n = 25) or without autapse (blue, n = 11) (two-sample Kolmogorov-Smirnov test D = 0.200 with P = 0.884, or Mann-Whitney U-test P = 0.836). Right: Plot shows lack of correlation between autapse strength (nS) and patient age (years). Red line depicts regression. (Spearman r = -0.142, P = 0.615, n = 14). We now mention these results in the manuscript.
